# Disordered Eating and Eating Disorders in Pediatric Obesity: Assessment and Next Steps

**DOI:** 10.3390/ijerph20176638

**Published:** 2023-08-24

**Authors:** Eileen Chaves, D. Thomas Jeffrey, Dominique R. Williams

**Affiliations:** 1Department of Pediatrics, The Ohio State University College of Medicine, Columbus, OH 43210, USA; 2Center of Healthy Weight and Nutrition, Nationwide Children’s Hospital, Columbus, OH 43205, USA; 3Psychology Department, The Ohio State University, Columbus, OH 43210, USA; jeffrey.60@buckeyemail.osu.edu; 4Abbott Nutrition, Columbus, OH 43219, USA; dominique.williams1@abbott.com

**Keywords:** disordered eating, eating disorders, pediatric obesity

## Abstract

While the exact prevalence of disordered eating in youth who are overweight and have obesity has not been determined, studies show that the odds of a young adult (18–24 years) with obesity engaging in disordered eating behaviors is 2.45 times more likely to occur than in young adults with Body Mass Indexes (BMI) in the normative range. The purpose of this review is to highlight the role that disordered eating and eating disorders may play in pediatric obesity and the importance of screening for these conditions. The ability to identify and assess disordered eating alters the course of treatment. Without an understanding of the intersection of obesity and disordered eating, medical providers may continue treatment-as-usual. Doing so may inadvertently contribute to internalized weight bias in patients with obesity and exacerbate their disordered eating symptoms and behaviors. In addition, understanding the spectrum of disordered eating in pediatric patients with obesity allows providers to tailor treatments, discuss food and physical activity differently, and know when to refer patients to eating-disorder-specific providers for continued treatment.

## 1. Introduction

Eating disorders (ED) are serious physical and mental illnesses that involve complicated and harmful relationships with food, eating, physical activity, and body image [1]. They affect people of all genders, age groups, ethnicities, body sizes, socioeconomic statuses, religions, and sex. EDs affect at least 9% of the population worldwide. In the United States, about 28.8 million Americans, or 9% of the population, will develop an eating disorder in their lifetime [2]. Fewer than 6% of people with an ED are medically diagnosed as “underweight” [3]. Having increased adiposity is a risk factor for developing an ED, including among culturally and linguistically diverse individuals and males [4,5,6]. Additionally, the development of becoming overweight or obese is also a common outcome for individuals with binge eating disorder (BED) and bulimia nervosa (BN). Despite this, people with increased adiposity are half as likely as individuals with average or below average BMIs to be diagnosed with an ED [7].

There is often confusion between disordered eating behaviors (DEBs) and EDs. DEBs include disordered eating (DE) as well as EDs. DE describes a range of observable eating behaviors that may cause distress, be maladaptive, and often complicate obesity treatment, while EDs refer to clinically diagnosable eating disorders. DE is usually a subclinical presentation of an ED and differs in terms of frequency and severity of the abnormal eating pattern. The purpose of this paper includes increasing awareness of the range of pre-existing DEBs and EDs in children and adolescents with obesity, understanding the importance of screening for these conditions, recognizing how DEBs and EDs complicate obesity treatment, and knowing when and how to refer children and adolescents for additional treatment. Research has shown that engagement in a structured weight management program is protective against EDs. However, because children and adolescents with obesity may engage in unhealthy weight practices to lose weight, it is important to identify pre-existing DEBs/EDs prior to the start of pediatric weight management to ensure that appropriate treatment is provided [8].

## 2. Disordered Eating vs. Eating Disorders

The distinction between disordered eating and eating disorders is one of degrees. While the terms disordered eating (DE) and eating disorders (ED) are often used interchangeably, the difference between the two is the frequency and severity of the behaviors observed. For the purpose of this review, the term disordered eating behaviors (DEBs) will be used to describe the *range* of disordered eating behaviors, including both subclinical presentations of eating disorders (often referred to as disordered eating) and eating disorders. See Figure 1.

There are different constructs that make up disordered eating, including a negative attitude towards weight and shape, unhealthy weight control behaviors, and binge eating. In contrast, eating disorders are defined as psychiatric conditions that result in deviant eating or weight control behaviors [9] and are diagnosed by health professionals using the diagnostic criteria in the DSM-5-TR. The constructs that make up disordered eating are also present in eating disorders, often making it difficult to distinguish between the two. However, the salient difference between disordered eating and eating disorders is the frequency and severity of the abnormal eating pattern. For instance, when diagnosed with binge eating disorder (BED), an individual must have episodes in which they consume a large amount of food while exhibiting a loss of control over eating at least once per week on average over the prior three months [10]. Additionally, the individual must meet at least three of the following features: eating at a faster pace than normal, eating until feeling uncomfortably full, eating large amounts in the absence of hunger, eating alone due to embarrassment of the amount of food consumed, and feeling disgusted, depressed, or guilty after a binge eating episode. Finally, this eating pattern must also cause distress and must not be followed by compensatory behaviors, such as purging [10]. This criterion does present an overlap of the constructs of disordered eating, specifically in regards to binge eating [11]. However, the requirements of the frequency in which the episodes occur and the severity of these episodes is what truly qualifies them as part of an eating disorder. For a comparison between ED diagnoses and subclinical presentations, see Table 1.

## 3. Prevalence of Obesity and DEBs

The frequency of DEBs and EDs in children and adolescents with obesity is unclear. However, a Canadian sample of 3043 adolescents found that the prevalence of estimated subclinical or full-threshold eating disorders was higher in adolescents with obesity (9.3% in males and 20.2% in females) when compared with their healthy weight peers (2.1% males and 8.4% females) [3]. According to the Centers for Disease Control and Prevention (CDC), obesity (BMI >95th percentile-for-age) affected approximately 14.7 million children and adolescents in 2020, with rates of 12.7% among 2-to-5-year-olds, 20.7% among 6-to-11-year-olds, and 22.2% among 12-to-19-year-olds [16]. The COVID-19 pandemic has exacerbated these statistics; in children aged 2-to-19-years-old, the rate of BMI increase approximately doubled during the pandemic compared to prepandemic, with a more significant increase seen in children who are overweight or obese [17]. Rates of DEBs are higher in children with obesity than children of normal weight [15]. In both boys and girls, the onset of eating disorders and disordered eating behaviors usually occurs between childhood and adolescence [5]. For boys, many male eating disorder patients report being overweight or having obesity during the developmental periods of childhood or adolescence [18]. Similar to obesity, childhood eating disorder diagnoses increased during the pandemic, with a 15.3% increase in the first year of the pandemic compared to the previous year [19]. When considering the different constructs of disordered eating, approximately 25% of adolescent girls who are overweight or have obesity overestimate their shape and weight. Likewise, 76% of girls and 55% of boys with obesity endorse unhealthy weight control behaviors, such as food restriction, ‘dieting’, irregular or inflexible eating patterns, or compulsive eating [15]. More than 25% of adolescents who are overweight or have obesity endorse binge eating episodes, with 30% reporting a loss of control (LOC) over eating [15].

## 4. Screening for DEBs

In a pediatric weight management population, treatment of children and adolescents with disordered eating and eating disorders is different from the typical standard of care. Typical standard of care for children and adolescents with obesity involves the following: supporting healthy eating behaviors, physical activity, and healthful behavior change [20]. For example, families are commonly counseled to change eating behaviors, like portion size or consumption of energy-dense/low-nutrient foods. Children and adolescents with obesity are also often advised to set goals and identify behavioral change targets (i.e., increase physical activity, achieve weight reduction). These recommendations can be problematic in children and adolescents with obesity who also present with disordered eating. In fact, rigid expectations for foods consumed, including describing foods as either “good” or “bad”/ “healthy” or “unhealthy”, can trigger increased food restriction, all-or-nothing thinking, and cutting out whole food groups [15]. Recommending a certain number of daily physical activities can trigger excessive or compensatory physical activity in youth with obesity and concomitant disordered eating behaviors [15,21]. It should be noted, however, that structured obesity treatment has been shown to be associated with reduced ED prevalence, ED risk, and ED symptoms [8].

Pediatric obesity treatment interventions should include screening for ED/DEB risk factors at pre- and postintervention and follow-up [22]. Timely recognition of DEBs in children and adolescents with obesity is critical to efforts to achieve improved health and quality of life. Medical providers play an important role in the identification of disordered eating behaviors and mitigation of symptoms that could develop into an eating disorder. Similarly, medical providers play a crucial role in screening for DEBs in children and adolescents with obesity. Identification of DEBs, management of related symptoms, and screening for EDs can help providers determine the treatment that best suites the needs of the patient. With the recent addition of antiobesity medications (AOMs) such as GLP-1 agonists, there are not yet data on the effects of AOMs on existing DEBs. This is an area for future research. Table 2 provides an overview of brief, validated screeners that can be used with children as young as 5 years old.

## 5. Effects of DEBs on Pediatric Obesity

### 5.1. Medical

Based on history, physical exam, and family history, children and adolescents with obesity need to be screened for cardiometabolic and obesity-related complications [22,23]. The effects of DEBs on pediatric obesity and overall health may vary [9]. For example, LOC or binge eating episodes characterized by palatable foods comprised of refined, processed carbohydrates and saturated fat can affect conditions like diabetes, hepatic steatosis, and dyslipidemia [24]. LOC eating is also associated with and predictive of psychological symptoms, high weight, and worsened cardiometabolic health [25]. On the other hand, rigid beliefs about food or restriction of food can lead to elimination of food groups. As a result, patients may experience malnutrition, fatigue, hair loss, sleep disturbance, and worsening mood [26,27]. Importantly, EDs carry one of the highest rates of mortality of any mental health condition [28]. Even with recent treatment advances, mortality rates are high for individuals diagnosed with anorexia nervosa (AN) and bulimia nervosa (BN). In those individuals who have received inpatient treatment, mortality risk is five times higher than for same-aged individuals without AN or BN [29].

Interestingly, there may also be a bidirectional relationship between sleep and disordered eating [28]. Insufficient sleep and untreated sleep disturbance (e.g., poor sleep hygiene, delayed sleep latency, obstructive sleep apnea, etc.) are associated with increased appetite and less restraint, eating in the absence of hunger, and preference for foods high in sugar and fat. Additionally, those with emotional eating may be more sensitive to sleep-related changes like altered mood and increased sensitivity to food as a reward [30]. Likewise, for some, eating helps to combat feelings of sleepiness or fatigue. Though the mechanisms for the relationship between sleep and disordered eating are not well understood, it is likely due in part to dysregulation of hunger/satiety hormones like ghrelin, insulin, etc. [31].

Disordered eating with compensatory behaviors like excessive exercise, self-induced vomiting, purging, or use of laxatives/diuretics may result in electrolyte disturbances with symptoms that vary in severity from mild fatigue to cardiac dysrhythmia [27]. However, DEBs typically differ from eating disorders in the frequency and severity of symptoms such that the risk for eating-related laboratory complications is low. Nonetheless, DEBs undermine obesity treatment by contributing to patterns of weight gain, avoidance of physical activity, or feelings of shame and guilt [9,24,27].

### 5.2. Psychological

DEBs contribute to a range of psychological complications in children and youth. These include disordered cognitions and self-perceptions [32], emotional dysregulation [33], feelings of shame and guilt [34], anxiety [35], depression [36], social withdrawal [37], and difficulty concentrating [38]. Specifically, LOC eating is related to both increased weight in children and adolescents as well as increased psychosocial impairment [39]. As a result, these psychological problems often contribute to and exacerbate DEBs, creating a cycle of negative thought patterns, which in turn can lead to emotional dysregulation. Emotional dysregulation often contributes to DEBs, as they are an attempt to cope with thoughts and feelings, and can then lead to an increased likelihood of becoming overweight or developing obesity. Developing an understanding of the interplay between the thoughts, feelings, behaviors, and associated psychological symptoms that contribute to DEBs aids providers in deciding when and how to intervene to disrupt this cycle [40]. Without this understanding, providers may unknowingly overlook mood and behavioral symptoms or view them as unrelated to being overweight and having obesity in youth.

### 5.3. Quality of Life

Both the medical and psychological effects of DEBs often affect an individual’s quality of life (QoL). The term quality of life (QoL) refers to an individual’s health, comfort, and happiness [41]. Fontaine and Barofsky (2001) define health-related quality of life (HRQoL) as a “multidimensional construct, encompassing emotional, physical, social and subjective feelings of well-being which reflect an individual’s subjective evaluation and reaction to health or illness” [42]. For some youth, obesity can affect not only mood but also sleep, motivation, physical mobility, and body image. Moreover, obesity can negatively impact interactions with peers and family members, including the ability to engage in tasks or experiences typical of same-aged peers. Finally, obesity increases the risk of developing cardiometabolic conditions, such as type 2 diabetes mellitus (T2D) [42,43]. Research shows an association between DEBs and impairments in HRQoL, specifically in the domain of emotional functioning in children and youth with obesity [44]. Unhealthy weight control behaviors are associated with perceived emotional well-being and specifically to emotional distress in children and youth with obesity. DEBs often function as a coping strategy to help modulate negative emotions [45].

## 6. Practical Next Steps

After identifying that a child or adolescent who is overweight or has obesity is also experiencing DEBs, it is important to know how this affects treatment and what next steps to take. Continuing with treatment as usual can result in both patient and provider frustration due the patient’s unintentional lack of adherence to obesity treatment. Having the expectation that a patient can engage in healthy lifestyle changes while also experiencing negative cognitions about their body weight/size, experiencing guilt and shame related to their eating habits, and engaging in either binge eating or food restriction is often unrealistic [46]. Multiple studies have shown that medical providers, specifically physicians, often feel unprepared to treat DEBS/EDs once they have been identified due to lack of training in this area [47,48]. In a qualitative study by Tse et al. (2022), physicians admitted feeling hesitant to approach the subject of EDs with their patients, allowing their own biases related to a patient’s body size/shape to influence their perceptions about the patient and their eating behaviors [48]. Physicians reported often overlooking or downplaying the information a patient shared related to their eating behaviors.

Interventions for obesity should focus on healthy body image and an overall increase in health-related quality of life [49]. Interventions for DEBs begin with increased provider awareness of and assessment of one’s own implicit biases related to being overweight and having obesity. When asking questions about a child’s or adolescents’ eating behaviors, it is helpful to always ask permission to discuss this topic and, if permission is not given, to respect the patient’s autonomy and discuss a different topic [50]. When permission is given to talk about eating behaviors and habits, a nonjudgmental conversation is more likely to occur. Nonjudgmental curiosity about an individual’s behaviors and thoughts helps to create more effective communication and allows children and adolescents to feel they will be heard and understood. Additionally, using the language children and adolescents use when describing their weight can help youth feel heard and understood during these conversations. Recent research has shown that this is particularly relevant when talking with Black and Latine youth, who report a preference for words such as “thick” and “curvy” to describe their weight [51,52]. Table 3 provides specific and practical language that can be used to both begin the conversation related to eating behaviors and begin treatment of challenging and changing DEBs.

## 7. Knowing When to Refer

Once a child or adolescent with obesity has screened positive for DEBs or an ED, knowing when and how to treat the patient as well as which type of treatment to refer is key. See Figure 2. Medical stability should first be determined. If a child or adolescent is medically stable, the ideal outpatient treatment team should include an experienced therapist or psychologist, a dietician, and a medical provider who is knowledgeable with both obesity treatment and DEBs [54]. Children and adolescents need to have regular, consistent appointments with their treatment team. The focus of treatment should be on helping the child or adolescent to fuel their bodies appropriately; establish regular meals and snacks throughout the day; increase emotional awareness; recognize and challenge negative, automatic thoughts related to body image, eating, and weight; and develop a more positive relationship with food and their body. Cognitive behavioral therapy (CBT) is often the first-line treatment of choice for both EDs with restrictive as well as binge eating patterns and can also be used to treat DEBs [54].

### Criteria for Medical Stability

Obesity medicine providers can ensure holistic care by addressing issues with sleep, recommending behavioral or pharmacotherapy for mood and behavior disorders, and advocating for the use of weight-neutral medications or finding alternatives to weight-promoting medications [24,55]. Children and adolescents with concomitant obesity and disordered eating behaviors can generally be managed in primary care or by the multidisciplinary weight management team [54]. If the DEBs become more frequent or there is evidence of an emerging eating disorder, then consider postponing obesity treatment in favor of higher acuity care or more time-sensitive medical treatment [27]. Table 4 summarizes medical complications that require referral to the emergency room or hospital admission. The Academy for Eating Disorders’ (AED) Guide to Medical Care is comprehensive and contains additional information related to medical and psychiatric symptoms indicating hospital admission [56].

## 8. Conclusions

The lifetime prevalence of ED is 2.2% in males and 4.93% in females. In individuals with obesity, the lifetime prevalence and past-12-month prevalence of ED is highest in individuals with class III obesity [57]. While the exact prevalence of DEBs is unknown in children and adolescents with obesity, the frequency at which these behaviors occur can affect the course of obesity treatment. Having an awareness that DEBs and obesity frequently co-occur is key for providers who treat pediatric obesity. Screening for DEBs with validated screeners in private locations helps to identify DEBs/EDs at the start of treatment, helping to shape obesity management, provide evidence-based care for both DEBs and obesity, and help providers know when to pause obesity treatment and refer patients for ED-specific treatment.

## Figures and Tables

**Figure 1 ijerph-20-06638-f001:**
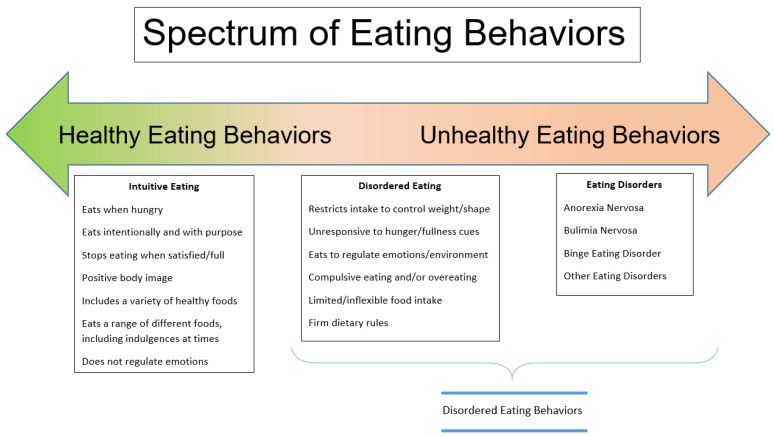
Eating Behaviors.

**Figure 2 ijerph-20-06638-f002:**
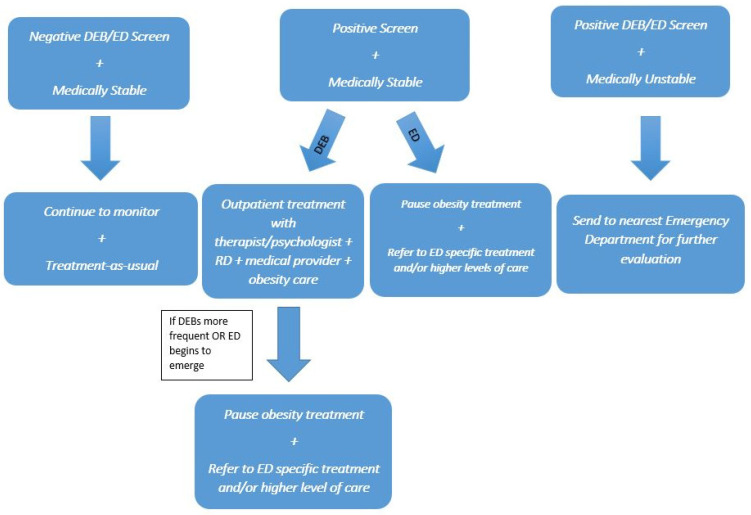
DEBs/ED: When to Refer for Additional Treatment.

**Table 1 ijerph-20-06638-t001:** Disordered Eating Behaviors (DEBs). Copyright © 2023 Roberts and Chaves. This is an open access article distributed under the terms of the Creative Commons Attribution License.

Disordered Eating Behaviors (DEBs) [1,12,13,14,15]		
Diagnosis	Eating Disorder Presentation	Disordered Eating Behaviors/Subclinical Presentations of ED
Anorexia Nervosa (AN)	Restriction of energy intake relative to requirements, leading to significantly low body weight [1] Intense fear of gaining weight or becoming fat even though at a significantly low weight Disturbance in the way in which one’s body weight or shape is experienced In children and adolescents, a BMI less than 85 percent of body weight expected for age and height or failure to gain weight during a growth period, leading to body weight less than 85 percent of that expected	Refusal to maintain body weight over a minimal normal weight AND/OR Intensive fear of gaining weight or becoming fat, even though underweight [12,14]
Atypical Anorexia Nervosa (AAN)	All criteria for AN are met, except that despite significant weight loss, the individual’s weight is within or above the normal range for BMI [1,8,9]	All criteria for subclinical AAN met, except that the individual’s weight is within or above the normal range for BMI [1]
Bulimia Nervosa (BN)	Recurrent episodes of binge eating (eating large amount of food in discrete period of time) + loss of control during binge eating [1] Recurrent compensatory behavior to prevent weight gain Self-evaluation is unduly influenced by body shape and weight Occurs at least once a week for 3 months	Recurrent episodes ≥2 times weekly of binge eating OR Loss of control during binge eating AND Compensatory behaviors of preventing weight gain [12]
Binge ED (BED)	Recurrent episodes of binge eating associated with 3 or more of the following: (1) Eating more rapidly than normal (2) Eating until uncomfortably full (3) Eating large amounts of food when not physically hungry (4) Eating alone due to embarrassment about how much one is eating. (5) Feeling disgusted with oneself, depressed, or guilty afterwards AND Marked distress regarding the binge eating No compensatory behaviors to prevent weight gain [1] Binge eating occurs on average at least once per week for 3 months	Having an episode of binge eating <2 times weekly [12]

**Table 2 ijerph-20-06638-t002:** Screeners *.

Screener Name:	Ages Validated:	# of Items:	Screens for:
Children’s Brief Binge Eating Questionnaire	7–18 year olds	7 items	Binge Eating Disorder
Child Binge Eating Disorder Scale	5–13 year olds	7 items	Binge Eating Disorder
Eating Disorders Screen for Primary Care (ESP)	18 years+	4 items	All Eating Disorders
Sick, Control, One, Fat, and Food (SCOFF)	11 years+	5 items	All Eating Disorders, including Atypical Anorexia Nervosa
Adolescent Binge Eating Disorder (ADO-BED) Questionnaire	12 years+	6 items	Binge Eating Disorder
Screen for Disordered Eating (SDE)	18 years+	5 items	All Eating Disorders
Eating Disorder Diagnostic Scale (EDDS)	13 years+	22 items	Anorexia Nervosa, Bulimia Nervosa, and Binge Eating Disorder
Eating Disorder Questionnaire—Short Form (EDE-QS)	14 years+	12 items	Anorexia Nervosa, Bulimia Nervosa, and Binge Eating Disorder

* Please note this is not an exhaustive list of all ED screeners; rather, it is a sampling of the most common screeners used to screen for ED in children and youth.

**Table 3 ijerph-20-06638-t003:** Practical Questions and Brief Interventions [53].

Goal	Questions/Statements
Start a conversation about eating habits	Would it be okay if we discussed your eating habits? May we discuss how you typically eat?
Assess motivation to change eating habits	On a scale of 0 to 10, how important is it for you to change your eating? What would make it more important? What do you like about the way you eat? What do you dislike? How would your life be different if you did not need to spend so much time thinking about your eating?
Determine the antecedents and consequences of disordered eating patterns	Do you ever feel that you lose control over the way you eat? How often does that happen? When are you most likely to binge? Sometimes people binge and purge when they feel emotional (sad, stressed, bored, worried, etc.); do any of those situations apply to you? How do you feel before you binge/purge? After you binge/purge? How does eating affect your ability to function during the day?
Develop alternatives to bingeing/restricting food	When you feel an urge to binge or restrict eating what could you do instead? Consider activities that you could do in the situations when you are most likely to binge or restrict food
Challenge negative thinking	Who demands that you must be perfect? Who determines how you think about yourself? What can you control?

**Table 4 ijerph-20-06638-t004:** Complications of ED that Require Immediate Attention.

Refusal to eat Signs of dehydration (i.e., dry mouth, sunken eyes, poor skin turgor, lethargy, infrequent voiding) Orthostatic hypotension Low heart rate (<50 bpm) Low body temperature (<35.5 degrees Celsius) Cardiac dysrhythmia or prolonged QT Active suicidal ideation

## Data Availability

Not applicable.

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
