# Peer review of "Disordered Eating and Eating Disorders in Pediatric Obesity: Assessment and Next Steps"

_ijerph, 2023, doi:10.3390/ijerph20176638_

Round 1

Reviewer 1 Report

Thank you for the opportunity to review this manuscript on an important topic. The authors discuss the overlap between obesity, disordered eating behaviors and eating disorders in children and adolescents. They provide a conceptualization of the overlap, discuss screening for eating disorders, provide a brief overview on the effects of eating disorders in children, and lastly provide recommendations for medical care. Although this topic is of the utmost importance, I have some concerns that ultimately reduce enthusiasm for this manuscript. 

Broadly, it appears a more comprehensive of the literature is needed to inform the conceptualization and recommendations that are provided. There are several relevant publications on these topics that were not referenced in the manuscript. I recommend the authors review work published on this topic by Hiba Jebeile, Marian Tanofsky-Kraff, and Dianne Neumark-Sztainer, to name a few. Moreover, throughout the manuscript there are several sentences that require citations. 

Prevalence

There are several published studies on the prevalence of disordered eating behaviors, and eating disorders in youth with overweight. For some examples see Byrne ME, LeMay-Russell S, Tanofsky-Kraff M. Curr Obes Rep. 2019 Mar;8(1):33-42. doi: 10.1007/s13679-019-0327-1. PMID: 30701372)

Conceptualization and treatment implications 

The rationale and utility of distinguishing between disordered eating behaviors and eating disorders could be made clearer. For example, if a person has all the clinical features of binge-eating disorder, but not the frequency, they would be diagnosed with an eating disorder, specifically OSFED- BED of limited frequency or duration. This is particularly relevant when assessing eating disorders in children and adolescents, as youth frequently report loss-of-control eating (a core feature of binge eating) but not consumption of an objectively large amount of food (required for a diagnosis of binge eating). This has led some to suggest different diagnostic criteria for pediatric binge eating disorders (see Tanofsky-Kraff M, Marcus MD, Yanovski SZ, Yanovski JA. Eat Behav. 2008 Aug;9(3):360-5. doi: 10.1016/j.eatbeh.2008.03.002. Epub 2008 Apr 7. PMID: 18549996; PMCID: PMC2495768.) 

The authors appear to use DEB and EDs interchangeably throughout the document, which detracts from the proposed distinction and its implications for treatment.  

Some of the behaviors in figure 1 listed as healthy can be unhealthy (e.g., allowing for indulgences) depending on the frequency and rigidity of the behavior. A more detailed consideration of these factors might help provide clarity regarding the author's conceptualization of when behaviors are or aren’t unhealthy. Moreover, body image and diet culture are not eating behaviors, therefore, their inclusion in figure 1 is somewhat confusing. 

Others have previously provided conceptualizations of the overlap between disordered eating behaviors and obesity, as well as the distinction between disordered eating behaviors and eating disorders. Referring to these conceptualizations (or others) might bolster support for the model proposed in the present paper.

-       Tanofsky-Kraff, M., Schvey, N. A., & Grilo, C. M. (2020). American Psychologist, 75(2), 189–203. https://doi.org/10.1037/amp0000592

-       Hayes JF, Fitzsimmons-Craft EE, Karam AM, Jakubiak J, Brown ML, Wilfley DE. Curr Obes Rep. 2018 Sep;7(3):235-246. doi: 10.1007/s13679-018-0316-9. PMID: 30069717; PMCID: PMC6098715.

-       Stabouli S, Erdine S, Suurorg L, JankauskienÄ— A, Lurbe E. 2021 Nov 29;13(12):4321. doi: 10.3390/nu13124321. PMID: 34959873; PMCID: PMC8705700.

-       Haynos, A.F., Field, A.E., Wilfley, D.E. and Tanofsky-Kraff, M. (2015). Int. J. Eat. Disord., 48: 362-366. https://doi.org/10.1002/eat.22355

I would argue that obesity treatment may need to be paused even in the case of some sub-threshold disordered eating behaviors. For example, if a child is self-inducing vomiting, that behavior should be addressed before moving forward with weight loss, even if the frequency is limited. 

Screening measures 

Table 2 – it is curious that the most well-validated and widely used screener is not reported in the table. Is there a reason the authors excluded the EDE-Q; Fairburn, C. G., & Beglin, S. J. (1994). International journal of eating disorders, 16(4), 363-370.

Related, there is a recently published review on measures to screen for eating disorders in youth with obesity that might be of relevance.  House ET, Lister NB, Seidler AL, Li H, Ong WY, McMaster CM, Paxton SJ, Jebeile H. Int J Eat Disord. 2022 Sep;55(9):1171-1193. doi: 10.1002/eat.23769. Epub 2022 Jul 9. PMID: 35809028; PMCID: PMC9545314.

Associated factors

Although the authors are correct that ED confers greater medical consequences than DEB, there is research showing that even sub-threshold DEB have medical consequences. Referencing these studies would bolster support for their argument that even disordered eating behaviors are relevant for pediatric obesity treatment. There are many studies, but for a recent example see Shank, L.M., Moursi, N.A. & Tanofsky-Kraff, M. Curr Diab Rep 22, 257–266 (2022). https://doi.org/10.1007/s11892-022-01466-z

There are recent reviews on the psychological consequences of disordered eating behaviors in youth. Citing these papers could bolster support for this section of the present paper. There are many examples, but one paper that comes to mind is Goldschmidt AB. Obes Rev. 2017 Apr;18(4):412-449. doi: 10.1111/obr.12491. Epub 2017 Feb 6. PMID: 28165655; PMCID: PMC5502406.

Citations for clinical symptoms warranting an immediate referral to the emergency room should be provided to ensure the list in the manuscript is comprehensive. Here is a link to the medical care guidelines put forth by the Academy for Eating Disorders. This list may not be designed for the context of pediatric obesity, but it is comprehensive and potentially relevant. https://higherlogicdownload.s3.amazonaws.com/AEDWEB/27a3b69a-8aae-45b2-a04c-2a078d02145d/UploadedImages/Publications_Slider/2120_AED_Medical_Care_4th_Ed_FINAL.pdf

Provider language

I greatly appreciated the section on how providers can approach discussions about eating in a less stigmatizing manner. I think this is a very important point to include. An additional recommendation that the authors might add is for the provider to use the language the patient uses to describe their eating. For example, not all patients will use the words binge and purge, but they might use other words to refer to these behaviors. Adopting the patient’s language can improve rapport and openness.

Reviewer 2 Report

Please see attached 

Reviewer 3 Report

Thank you for taking the time to write and submit this incredibly thoughtful and well-written review. This topic is so timely given the recent AAP CPGs on Pediatric Obesity, and provides practical guidance to those who are concerned about "causing" or exacerbating disordered eating or an eating disorder with obesity or weight management treatment. I have a few minor suggestions, but overall very much appreciate the figures and tables, which are practical and easy to follow.

Introduction:

I think that the very loud concern out there is that obesity treatment "causes eating disorders"  and therefore really highlighting the point that the Cardel et al. (reference #16) article makes there is a difference between self-directed diets and evidence-based  weight management  in the introduction is important.  If there is space, I would also directly cite articles that show improvement in disordered eating symptoms with structured obesity treatment with a dietary component (Gow et al, 2019) or even improvement in QoL treatment with semaglutide (Weghuber 2022). 

Line 42-43: To this point, I would just add the words "pre-existing" before DEBs or EDS in the paper objective statement. 

Page 4; "Screening for DEBs": Understanding that the use of anti-obesity medications is pediatric obesity is very new, have the authors found any data on the use of AOMs or bariatric surgery triggering medication overuse or disordered eating? Even a statement like "there is not yet data on the effects of AOMS on existing DEBS" may be a worthwhile addition, given that AOMS or bariatric surgery can be part of a multi-component obesity management approach. 

Round 2

Reviewer 2 Report

The authors did not provide any reason for not addressing one of the comments, which asked for a focus on interventions for individuals with overweight or obesity in maintaining a healthy weight status. 

If Table 2 does not include a comprehensive list of all screeners, this should be explicitly mentioned in the manuscript and as a footnote beneath Table 2.

The introduction section could benefit from streamlining and should be reviewed by a native English speaker, as there were multiple instances of run-on sentences. 

It might be more appropriate to move the statement "There is often confusion between disordered eating behaviors and eating disorders" from line 38 to section 2.0 

The paper does not clearly state the rationale behind selecting pediatrics and children as the focal point of this paper. 

Overall, this paper may benefit from editing by a native English speaker or a professional editor to improve the flow of this paper. 
